## Research Article

adolescents; digital mental health; low- and middle-income countries; mixed-methods; mobile health

**Corresponding author:**
Sonto Gugu Madonsela;
Email: sonto.madonsela@wits.ac.za

# Adolescent readiness for mobile mental health support in Soweto: A mixed-methods study

Sonto Gugu Madonsela[1] 🔗, Jennifer Watermeyer[1], Lisa Jayne Ware[2] and Megan Scott[1]

[1]Health Communication Research Unit, School of Human and Community Development, University of the Witwatersrand Johannesburg, South Africa and [2]SAMRC/Wits Developmental Pathways for Health Research Unit, Department of Paediatrics and Child Health, School of Clinical Medicine, Faculty of Health Sciences, University of the Witwatersrand, Johannesburg, South Africa

## Abstract

Poor mental health is a growing issue among adolescents, with untreated conditions persisting into adulthood and typically increasing in severity. South Africa's mental health legislation faces key barriers to implementation due to limited access to treatment and support, as well as persistent challenges related to stigma, privacy concerns and affordability. Mobile mental health (M-mHealth) could be a sustainable and scalable alternative for reducing unmet needs for psychological services. This study aims to explore adolescents' perceptions, attitudes and intentions regarding M-mHealth interventions. The study involved two phases and used an explanatory sequential mixed-methods approach. 71 adolescents completed the survey in phase 1, while 56 adolescents participated in 9 focus group discussions in phase 2. Qualitative and quantitative data were analysed using thematic and descriptive analyses, respectively. Findings from both phases were integrated using the pillar integration process. Findings show that adolescents have a limited understanding of the broader concept of mental health, and stigma persists through the use of terms like "crazy" and "bewitched." Adolescents view M-mHealth positively because of its low cost, convenience and privacy. However, issues like data costs, smartphone affordability, and limited privacy at home could hinder its use. M-mHealth extends beyond the health sector and is constrained by infrastructural and socio-cultural barriers, including privacy concerns, high data costs, and stigma.

## Impact statement

Mobile mental health (M-mHealth) services have the potential to overcome barriers to mental health care and reduce the unmet needs for psychological services in LMICs. If M-mHealth will be useful in providing this support to adolescents in LMICs, it is critical to understand their perceptions, attitudes, potential facilitators, barriers and intentions for using M-mHealth. Using a mixed-methods design, this article demonstrates that while adolescents may recognise the potential benefits of M-mHealth interventions, there are potential challenges to consider. To be effective, M-mHealth interventions must be tailored to the specific context, taking into account its unique challenges and nuances.

## Introduction

Poor mental health and substance use disorders are among the leading contributors to years lived with disability (YLDs) among children and youth up to the age of 24 years globally, making up approximately 25% of YLDs within this age cohort (Sorsdahl et al., 2021). In South Africa (SA), more than one in 10 children has a diagnosable and treatable mental disorder, which includes depression, anxiety, post-traumatic stress disorder, conduct, learning and substance-use disorders, among others (Tomlinson et al., 2022). The risk of presenting with a mental condition is even greater for adolescents exposed to high levels of adversity, such as poverty, violence and sexual abuse (Babatunde et al., 2020). The substantial burden of adolescent mental health problems, coupled with limited treatment capacity, highlights the necessity of prevention approaches that reduce incidence and delay the onset of mental disorders.

mHealth for mental health (M-mHealth) has gained interest in SA, especially among adolescents, because of its potential to tackle adolescent mental health issues (Mindu et al., 2023; Dallison et al., 2025; Moffett et al., 2025). Despite many adolescents facing mental health issues, about 90% of children with mental disorders in SA cannot access care due to socio-economic challenges and low prioritisation by the government (Docrat et al., 2019; Babatunde et al., 2021; Tomlinson et al., 2022). The National Mental Health Policy Framework (2023–2030) emphasises the

government's commitment to children's mental health, but funding remains limited (Sorsdahl et al., 2023). It is therefore essential to develop sustainable and scalable care-delivery methods, particularly for high-risk groups such as adolescents.

While there is a growing body of research on adolescent M-mHealth in high-income countries, evidence-based research in low- and middle-income countries (LMICs) remains limited. A review of adolescent M-mHealth in LMICs found six studies from five countries. Results indicate that M-mHealth is promising but highlight the need for adolescent involvement in its design and development (Madonsela et al., 2023). While M-mHealth interventions have potential benefits, evidence for their long-term effectiveness is limited and has yielded mixed results, with other interventions reporting high dropout rates (Lehtimaki et al., 2021; Hall et al., 2022; Fernández-Batanero et al., 2025). A behavioural activation digital intervention incorporating gamification and peer support for adolescent depression in rural South Africa showed high acceptability of the Kuamsha app, supplemented by peer mentor calls (Moffett et al., 2025). This study indicates that technology alone does not ensure M-mHealth effectiveness; instead, contextual factors play a more crucial role. Similarly, despite broad acceptability, persistent engagement challenges with M-mHealth underscore the pressing need to address adaptation, co-design and feasibility in LMICs (Bear et al., 2022, 2024). Therefore, further research is needed on M-mHealth interventions targeted at adolescents in SA, including understanding their perceptions, attitudes, intentions to use and potential barriers within their context, in order to design M-mHealth interventions that are not only effective but also encourage ongoing engagement among users. Accordingly, this study had two main objectives: (i) to assess adolescents' mental health literacy in Soweto and (ii) to explore their perceptions, attitudes and intentions to use M-mHealth interventions.

This study was conducted with adolescents exposed to elevated levels of adverse childhood experiences in an LMIC country experiencing a significant demographic dividend. The research responds to the urgent need for scalable and sustainable approaches to support adolescent mental well-being. While mHealth interventions are expanding rapidly in LMICs, their application to mental health remains limited and frequently undermined by implementation and sustainability challenges. By examining these challenges, this study provides practical, context-specific guidance to inform the design, implementation and long-term viability of M-mHealth interventions, thereby strengthening their potential for real-world impact in similar settings.

## Methods

This study used an explanatory sequential mixed-methods design, beginning with quantitative data collection, followed by qualitative data collection to expand our understanding of the topic and inform interpretation of the quantitative findings. Explanatory sequential designs proceed in two distinct phases: first, collecting and analysing quantitative data, followed by qualitative data to explain or expand on the quantitative findings (Creswell and Plano Clark, 2018). The quantitative phase involved a survey completed by 71 adolescents aged 14–19 years, and the qualitative phase involved nine focus group discussions (FGDs) with 56 adolescents. To integrate the data, this study used the display format PIP, "Pillar Integration Process" to combine the quantitative and qualitative findings (Johnson et al., 2019).

### Study setting and participants

Ethical approval for this study was provided by the University of the Witwatersrand Human Research Ethics Committee (H22/10/30). Both phases were conducted in a peri-urban township in Johannesburg, South Africa, where one-third of the youth show depressive symptoms associated with high levels of substance abuse, incarceration, personal violence and early pregnancy (Kim et al., 2023). SM recruited adolescents at a centrally located youth development centre that supports after-school activities between March and October 2023. To be eligible to participate, participants had to be aged 13–19 years, reside in the area and participate in one of the programmes offered at the youth centre. Adolescents who were without parental consent were excluded. Adolescents received an information sheet and consent forms for guardian permission and their assent before completing the survey. Those interested who completed the survey were asked to provide contact details for FGDs.

### Study procedures and measures

The phase 1 quantitative data was collected and managed using REDCap electronic data capture tools (Harris et al., 2009, 2019). Mental health literacy was assessed using the "Mental Health Literacy Measure" by Jung and colleagues, which had previously shown good psychometric properties and construct validity among participants (n = 211) in a low-income setting (Jung et al., 2016). Reliability in this study was assessed with Cronbach's alpha, yielding a score above 0.7, indicating good internal consistency.

The Technology Acceptance Model (TAM), developed by Fred Davis in 1989, predicts how users accept new technology based on two main factors: (i) Perceived Usefulness (PU), which is the belief that technology enhances job performance and (ii) Perceived Ease of Use (PEOU), the belief that using the technology is effortless. These factors shape attitudes towards the technology, influencing behavioural intention and actual usage (Davis and Granić, 2024). TAM, adapted from a study by Ghani et al. (2019), was used to investigate adolescents' acceptance of M-mHealth to support their mental health needs. The adapted TAM focused on three related constructs, Perceived Usefulness, Behavioural Intention to Use and Attitude. In this study, the average Cronbach's alpha was lower for the behaviour construct, while the other constructs had alphas of 0.7. The reliability ranged from 0.5 to 0.6 across all constructs, indicating weak reliability. The survey was supplemented with a demographic questionnaire and additional items to examine smartphone ownership, use and internet access.

The quantitative results in phase 1 informed the development of the FGD guide for Phase 2. Together with a trained, experienced and multi-lingual research assistant, nine FGDs were conducted in English, IsiZulu and Sesotho. Each FGD lasted 60 to 90 min. Data collection continued iteratively until interviews yielded no additional insights, indicating saturation. Coding was conducted iteratively, with codes refined throughout analysis as patterns emerged. Coding decisions were reviewed collaboratively with co-authors, who provided ongoing feedback throughout the process. Audio recordings of the FGDs were transcribed verbatim, translated by multi-lingual transcribers and cross-checked by SM.

### Data analysis

Knowledge of mental health, perceived usefulness, attitudes and intention to use M-mHealth were measured on a five-point Likert

scale. Reliability for each item was assessed using Cronbach's alpha, with a score of 0.7 considered the threshold for reliability. Knowledge, perceived usefulness, attitude and behaviour (intention to use) items were summed to yield total scores for each construct. Negatively asked questions were reversed in their scoring.

Qualitative data were analysed using inductive thematic analysis (Braun and Clarke, 2023). Before coding, SM read and reread all transcripts to gain a broad understanding of the data and to develop a codebook. SM then submitted the codebook to the research team for review and approval. Once approved, SM generated the themes and submitted them to the research team for review and approval.

### Trustworthiness

In this study, trustworthiness was achieved through multiple strategies, guided by Shenton (2004) recommendations. Credibility was achieved through SM's familiarity with the participants, gained by visiting the centre and engaging with them prior to data collection. Probes were used to elicit detailed data and iterative questioning by returning to matters previously raised by participants and extracting related data through rephrased questions. Credibility was also enhanced by triangulating data from various stakeholders (via surveys and FGDs) and by collecting data through multiple methods (Shenton, 2004).

### Researcher positionality

The research team has interdisciplinary training in psychology, public health and genetic counselling, with the lead author primarily responsible for qualitative data collection and analysis. The team's prior experience working in this community with young people and conducting research in this context informed the framing of the research questions and the initial analytical lens. To engage reflexively with these perspectives, coding decisions and interpretations were reviewed collaboratively, with notes taken to document evolving interpretations. Quantitative analyses followed standardised procedures.

To integrate the quantitative and qualitative findings, the four stages of PIP were followed (Johnson, 2019). See the diagram in Figure 1 for the PIP process.

1. *Listing*: The joint display included raw data (e.g., percentages, selected quotations) and coded or grouped data that the researchers considered important for inclusion in the integration.

2. *Matching*: After including the relevant data in the QUANT columns, researchers aligned and refined the corresponding data in the opposite columns, organising the categories listed.

3. *Checking*: After matching the data and confirming the accuracy of the match, the researchers checked the data for quality purposes.

4. *Pillar building*: This stage involved comparing findings from the listing, matching and checking phases. It focused on integrating quantitative and qualitative data to derive insights and identify patterns, themes and possible explanations found in the pillar column.

### Results

This section provides the results of each phase of the explanatory sequential mixed-methods study. The findings offer a detailed understanding of M-mHealth for adolescent mental health in Soweto.

### Quantitative findings

Phone ownership was investigated to evaluate potential access to M-mHealth. Our results in Table 1 showed that a third of adolescents share phones with family members, and a quarter do not own their own phones, which may affect private access to M-mHealth tools. However, at least half reported that their family does not access their phone, and three-quarters own their device, indicating that, for many adolescents, private access to M-mHealth tools may be feasible, with no significant differences observed by gender.

Table 2 shows that many participants are regular phone users, indicating that smartphones are essential for their daily activities. Frequent interaction with digital platforms suggests a reliance on mobile connectivity, while less than half have internet access at home. Over half of the participants rely on mobile data or other means for internet access, which may hamper their use of M-mHealth tools. Only a third find it easy to access Wi-Fi, indicating that many struggle with connectivity. Despite its availability, only about a quarter of participants reported frequent use.

Table 3 shows the descriptive statistics for knowledge-oriented mental health literacy. Item 3 (72.73% agree) and item 5 (58.33% agree, 26.67% strongly agree) show high levels of agreement. These indicate that adolescents generally understand the benefits of early intervention and counselling in mental health care. Other items, such as item 11, show a spread of responses, indicating mixed

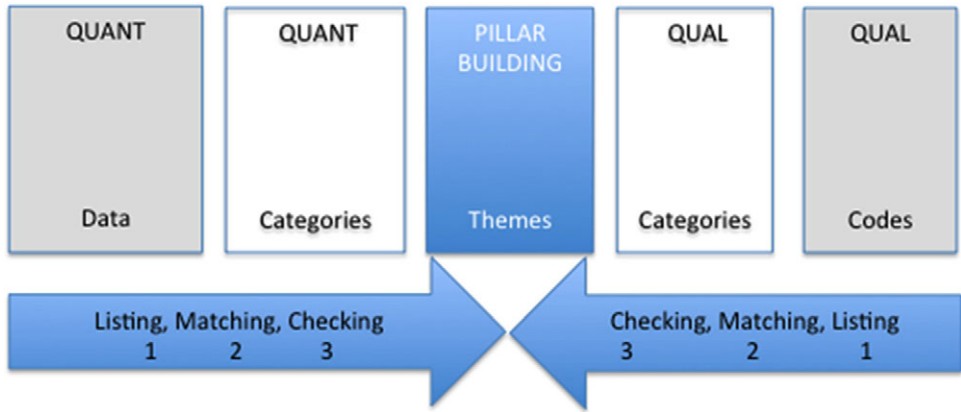

**Figure 1.** Pillar integration process.

**Table 1.** Phone ownership, sharing and family access to adolescent phones

|  | Total n(%) | Male n(%) | Female n(%) | P-value |
|---|---|---|---|---|
| Share phone with family |  |  |  |  |
| No | 45(68) | 16(64) | 29(70) | 0.569 |
| Yes | 21(32) | 9(36) | 12(30) |  |
| Can family access your phone |  |  |  |  |
| No | 36(55) | 15(60) | 21(51) | 0.487 |
| Yes | 30(45) | 10(40) | 20(49) |  |
| Phone ownership |  |  |  |  |
| No | 16(24) | 5(20) | 11(27) | 0.530 |
| Yes | 50(76) | 20(80) | 30(73) |  |

**Table 2.** Phone use and access to the internet

| Characteristics | Groups | Frequencies | Percentages |
|---|---|---|---|
| How often do you use your smartphone device? | 0 = less than once a week | 4 | 6.3 |
|  | 1 = once a week | 6 | 9.5 |
|  | 2 = several times a week | 6 | 9.5 |
|  | 3 = every day of the week | 20 | 31.8 |
|  | 4 = several times a day | 27 | 42.9 |
| How do you access the internet? | 0 = Mobile data | 25 | 25.2 |
|  | 1 = home Wi-Fi/fibre | 31 | 43.7 |
|  | 2 = School wi-fi | 9 | 12.7 |
|  | 3 = Free community Wi-Fi hotspot | 7 | 9.9 |
| How easy is it to access the internet? | 0 = impossible | 1 | 1.5 |
|  | 1 = difficult | 11 | 16.7 |
|  | 2 = somewhat difficult | 12 | 18.2 |
|  | 3 = easy | 21 | 31.8 |
|  | 4 = very easy | 21 | 31.8 |
| How often do you connect to the internet? | 0 = less than once a week | 17 | 25.8 |
|  | 1 = once a week | 10 | 15.2 |
|  | 2 = several times a week | 8 | 12.1 |
|  | 3 = every day of the week | 14 | 21.2 |
|  | 4 = several times a day | 17 | 25.7 |
| Use your mobile phone for | Texting (SMSs & WhatsApp) | 36 | 50.7 |
|  | Social media | 42 | 59.2 |
|  | Access the internet | 40 | 56.34 |
|  | Download apps | 28 | 39.44 |
|  | Listen to music | 35 | 49.3 |

understanding or uncertainty about certain symptoms. Items like item 9 reveal lower agreement, which may suggest a gap in understanding specific symptoms associated with mental health conditions.

Table 4 shows the perceived usefulness, attitude and intention to use M-mHealth interventions. Most participants agreed that mobile phones are convenient for mental health services via calls, SMS and WhatsApp. They felt it reduced delays in seeking help, showing a positive perception of M-mHealth. The reliability score for the perception construct was at least 0.6. The participants' attitudes towards M-mHealth were also assessed. Most participants viewed using their mobile phone for mental health positively, agreeing that M-mHealth would help them manage their mental health and preferring mobile options over in-person visits.

However, less than half of the participants were neutral about whether they would be embarrassed to access mental health services using their phones. The reliability score for this construct was at least 0.6. In general, participants' attitudes towards M-mHealth were positive. The participant's intention to use was assessed. Most participants expressed openness to using mobile phones for mental health support and felt comfortable discussing their issues with a counsellor via phone. The reliability score for this construct was 0.5.

### Qualitative findings

The qualitative findings were divided into three major themes: (1) adolescents understand mental health in terms of severe mental illness, (2) adolescents' perceptions and attitudes towards M-mHealth and (3) potential M-mHealth challenges. We discuss each major theme using illustrative quotes, focusing on sub-themes for each.

#### Adolescents describe mental health in terms of severe mental illness

Adolescents expressed that they have a limited understanding of mental health and struggled to conceptualise this construct. During the FGDs, when asked what mental health is, adolescents described it as being unhappy, having problems, being stressed and over-thinking.

> *It is when you are unhappy for a long time that it becomes a habit (FGD009, PT0, M).*

> *Maybe they'd be having a lot of problems, and they can't seem to know how to fix them. Or stress, they overthink (FGD006, P1, M).*

Upon further probing, adolescents described mental health exclusively as extreme psychiatric disorders and not day-to-day mental health struggles such as mild depression and anxiety. What is also apparent is the adolescents distancing themselves from those with mental disorders, "us vs. them."

> *I take them as someone who is dangerous because they are someone who is not mentally okay in the brain…So, he has to be in an environment that deals with people who have similar issues (FGD007, P2, M).*

**Talking about mental health.** Adolescents report discussing mental health in their communities, but often use negative terms like "crazy" to describe those with mental health issues. Additionally, they tend not to talk about everyday challenges, such as stress and anxiety.

> *…we sometimes talk about it [mental health], but we only talk bad things about it, we hardly talk positive things about it, we rather sometimes say someone is crazy, and they are not okay in the head, or we say the person is bewitched…(FGD007, P2, M).*

> *We do. But then we don't talk about it fully. We only do shortcuts. Um, for example, talking about, maybe someone, saying a person acts crazy or something in a bad way…Saying people are crazy (FGD003, P2, F).*

Adolescents said stigma and shame are some of the reasons why people do not speak about mental health, for fear of being judged or isolated from their communities.

> *Ehm……. It's because of being ashamed. People are scared of being laughed at, being a laughing stock in society…(FGD008, P2, M).*

**Table 3.** Knowledge-oriented mental health literacy

| | Knowledge items | n | 1 = strongly disagree | 2 = disagree | 3 = neutral/ I do not know | 4 = agree | 5 = strongly agree |
|---|---|---|---|---|---|---|---|
| 1 | Painful body, feeling tired and not eating can be a sign of depression | 59 | 7(11.86) | 6(10.17) | 10(16.96) | 29(49.15) | 7(11.86) |
| 2 | A person with a mental illness such as depression may see things that are not there | 53 | 3(5.66) | 13(24.53) | 6(11.32) | 23(43.4) | 8(15.09) |
| 3 | Early identification of symptoms of mental illness can improve the chances of getting better | 55 | 2(3.64) | 2(3.64) | 3(5.44) | 40(72.73) | 8(14.55) |
| 4 | Attending peer support groups or speaking to a social worker or psychologist helps people recover from mental illness | 62 | 1(1.61) | 3(4.84) | 6(9.68) | 29(46.77) | 23(37.1) |
| 5 | Counselling is a helpful treatment for depression | 60 | 0 | 2(3.33) | 7(11.67) | 35(58.33) | 16(26.67) |
| 6 | Going to therapy (like speaking to a social worker) can change the way a person thinks and reacts to pressure and stress | 54 | 1(1.84) | 3(5.56) | 4(7.41) | 30(55.56) | 16(29.63) |
| 7 | A person with bipolar disorder may show a big change in mood (switch from very happy to very sad) | 55 | 1(1.82) | 3(5.45) | 7(12.73) | 33(60.0) | 11(20.0) |
| 8 | Taking prescribed medications or treatment from the clinic or hospital for mental illness is helpful | 54 | 4(7.41) | 5(9.26) | 9(16.67) | 24(44.44) | 12(22.22) |
| 9 | When a person stops taking care of his or her appearance, it may be a sign of depression | 55 | 4(7.27) | 13(23.64) | 7(12.72) | 23(41.82) | 8(14.55) |
| 10 | Drinking alcohol makes symptoms of mental illness worse | 52 | 3(5.77) | 6(11.54) | 6(11.54) | 28(53.85) | 9(17.30) |
| 11 | A person with mental illness can receive treatment in a community healthcare setting (Clinic or hospital) | 59 | 2(3.38) | 7(11.89) | 9(15.24) | 31(52.54) | 10(16.95) |
| 12 | A person with an anxiety disorder has too much fear | 45 | 2(4.44) | 1(2.22) | 4(8.89) | 27(60.1) | 11(24.44) |

**Table 4.** Technology Acceptance Model (TAM)

| Construct | Items | n | 1 = strongly disagree | 2 = disagree | 3 = neutral/ I do not know | 4 = agree | 5 = strongly agree |
|---|---|---|---|---|---|---|---|
| Perceived Usefulness | It would be easy for me to use my mobile phone for mental health services | 64 | 2(3.13) | 2(3.13) | 5(7.81) | 38(59.38) | 17(26.56) |
| | Using my mobile phone (call, SMS, WhatsApp chatting, Mobile app) for mental health services would be useful for my mental health and wellbeing | 64 | 4(6.25) | 5(7.81) | 8(12.5) | 39(60.94) | 8(12.5) |
| | Using my mobile phone to speak with someone (Phone call or chatting on WhatsApp) about my mental health struggles would be useful | 64 | 3(4.69) | 2(3.13) | 7(10.94) | 37(57.81) | 15(23.44) |
| | Using my mobile phone for my mental health would allow me to deal with my mental health before it is too late | 63 | 2(3.17) | 7(11.11) | 3(4.76) | 34(53.97) | 17(26.98) |
| Attitude | Using my mobile phone for mental health needs is a good idea | 65 | 1(1.54) | 1(1.54) | 9(13.85) | 39(60.0) | 15(23.08) |
| | I feel positive/good about using my mobile phone for my mental health | 65 | 1(1.54) | 3(4.62) | 14(21.54) | 38(58.46) | 9(13.85) |
| | I believe that the use of mhealth for mental health services will help me take control of my mental health needs | 65 | 1(1.54) | 3(4.62) | 6(9.23) | 40(61.54) | 15(23.08) |
| | I generally favour the use of mobile phones for mental health services over going to the clinic/hospital | 65 | 1(1.54) | 10(15.38) | 7(10.77) | 33(50.77) | 14(21.54) |
| | I would be embarrassed if my friends saw me using my phone to access mental health services | 65 | 15(23.08) | 17(26.15) | 11(16.92) | 19(29.23) | 3(4.62) |
| Intention to use | I plan to frequently use my mobile phone for my mental health needs | 64 | 2(3.13) | 2(3.13) | 12(18.75) | 38(59.38) | 10(15.63) |

"We can't define it [mental health]," was predominant in adolescents' responses, highlighting their struggle to grasp this complex concept. While they may discuss mental health, they find it difficult to pinpoint a clear definition.

*We need to start by defining what mental health is. Maybe we talk about it, but we are not aware… Because you will find that we talk about it all the time when we are with the guys, but we see it in another way. (FGD008, P2, M).*

*I think that we always talk about the topic of mental health but we can't define it…what is mental health? We are always talking about it, but we will never be able to define it (FGD008, P3, M).*

### Parents/guardians shape adolescent mental health

Discussions highlighted the parental impact on adolescents' mental health. Participants described feeling unheard, misunderstood and pressured by their parents. Quotes reveal the complex dynamics between parents and adolescents regarding mental health.

*Again, the parents are the ones killing us a lot, mam…Yah, so in that moment you decided to go out to, to avoid the abuse. And when you tell them that they're not treating you right, yeah, they say you're disrespectful (FGD001, P4, M).*

*So the problem, our parents don't want to listen because there were no such things during their time. It was there but they didn't look into it to find out how it affects people and how do people feel. So, I think they have this thing that it wasn't there in the olden days, why is it starting now in the 2000s (FGD005, P3, F).*

### Adolescents' perceptions and attitudes towards M-mHealth

Participants had a positive view of M-mHealth for their mental health needs, believing it could enhance access to care through privacy, convenience and cost savings. They also mentioned using technology to search for solutions to mental health challenges.

*…For me, there's the internet you can use for research and check how you can help yourself. Such things --- and there's another one that looks like a robot, it's a new thing. You text it on WhatsApp. You can ask it anything and you will get a reply within five seconds and the answers are all right (FGD009, P0, M).*

### Potential M-mHealth benefits

Adolescents recognise the benefits of M-mHealth, valuing discreet access to mental health support. They see it as a convenient and cost-effective alternative to traditional services, which often involve additional expenses like transport.

*…unlike travelling from one place to another you can call them anytime to tell them how you feel. Imagine you are going through this and your appointment is next week. So you will have anger. At least over the phone, you can call that person that specific day… (FGD005, P0, F).*

*The advantages are that a person I don't know won't talk about it to other people. You can tell them how you feel, and then they will not pass it to the next person (FGD008, P3, F).*

*Like you don't use a taxi to go to ****. When they call you nothing happens, you just go there for nothing then they call you again. You wait for like an hour before they call you…to go there you need to wake up early like round 07h00 and you come back at like 18h00. With technology, you can log in at 07h00 and by 10h00, you are done (FGD007, P2, M).*

### Potential M-mHealth challenges

While adolescents expressed positive attitudes towards M-mHealth and acknowledged its benefits, they also expressed concerns about its potential challenges. Adolescents face barriers such as limited Wi-Fi access, insufficient mobile data and a lack of ownership of smartphones, which can hinder the use of M-mHealth. Safety and security, including the risk of crime or theft at the community-free Wi-Fi hotspots, were also significant concerns for adolescents.

**M-mHealth data costs and wi-fi connection.** With high data costs and challenges with the free community Wi-Fi, such as theft of mobile devices or the location being too far, it will be challenging for adolescents to engage fully with M-mHealth interventions.

*I am saying they can create an app, but most apps need you to have data to make use of it. There are people who can't afford those things. You will find that there is no Wi-Fi where they live, and they don't even have money to buy data. So, it's better to create a free app. There are people who do not have phones and those people will go through the very same problem of not being able to access the app (FGD009, P0, M).*

*Maybe at the clinic cause there was a place like in ****** you see, it's a school and it had free Wi-Fi and they had to change it because people were stealing. So it's not simple to get free WIFI. In some places the park is too far (FGD005, P0, F).*

**Mobile phones are viewed negatively in the home.** Guardians' oversight of adolescents' cell phones creates significant barriers to using M-mHealth. Adolescents are concerned about privacy invasion and report that their phones are often confiscated as punishment, limiting their access, and could hinder M-mHealth usage.

*You get that parents don't have money to buy the cell phones or parents just don't want to buy them because they are thinking cell phones are a bad influence in your life. (FGD005, P0, F).*

*…they[parents] have many questions and they told me if I get a phone I won't do well at school. It's not easy to get a phone with strict parents. (FGD005, P3, F)*

*You know, they are able to see who you talk to, what you search for, pictures, whose pictures are they. Everything. (FGD004, P3, F)*

### Integration of quantitative and qualitative findings

The PIP process resulted in the emergence of seven pillars from the integrated quantitative and qualitative results. The seven pillars are described below.

### Pillar one – A need for balance between ownership and surveillance

The quantitative findings indicate that adolescents own mobile phones, but some of their family members have access to their phones. This finding aligns with the qualitative findings, suggesting that guardians monitor their adolescents' phones and that phones are viewed negatively in the home.

### Pillar two – Internet access and financial constraints

Although most adolescents own mobile phones, as shown in the quantitative findings, the qualitative findings suggest that many struggle with affordability. The findings reveal financial challenges for some adolescents.

### Pillar three – Understanding mental illness through signs and symptoms

In the survey data, adolescents were able to recognise the signs and symptoms of mental illness. This was also evident in the qualitative findings, as adolescents described their understanding of mental health using signs and symptoms, such as being 'unhappy for a long time'.

### Pillar four – Guided recognition to identify mental illness

Adolescents scored highly for knowledge-oriented mental health literacy. On the contrary, the qualitative results show that adolescents struggle to conceptualise or describe the concept of mental health.

They also mentioned that they discuss it, but might not recognise it as such.

### Pillar five – Underutilised mental health support

While adolescents are aware that help for mental illness is available at local clinics, as per the quantitative findings, the qualitative findings revealed that they are reluctant to seek help at these clinics due to negative treatment from the staff. This integration of data indicates that, despite being aware of the availability of help at the clinic, adolescents are deterred from seeking assistance.

### Pillar six – Perceptions of M-mHealth

In the quantitative findings, most adolescents agreed that using mobile phones for mental health services would be easy and could help avoid delays in seeking help. Similarly, in the qualitative findings, adolescents mentioned benefits such as privacy and the absence of travel costs, which they found helpful.

### Pillar seven – Attitudes and intentions towards adoption of M-mHealth

In the quantitative findings, most adolescents agreed with the idea of using their mobile phones for mental health services, demonstrating a positive attitude towards M-mHealth. This was further emphasised in the qualitative findings, in which adolescents indicated that M-mHealth was their preferred method of seeking help compared with traditional avenues, which often entail travel expenses and long waiting times. The quantitative data indicate that adolescents would consider using their mobile phones for mental health services. The same sentiments were evident in the qualitative data, in which adolescents reported that they would use their mobile phones for mental health services and described various ways they could do so.

## Discussion

The present study contributes to the emerging field of M-mHealth in LMICs, which has limited research and remains dominated by research from the Global North. While M-mHealth might be useful to some extent, our findings show that M-mHealth extends beyond the health sector and is constrained by infrastructural and sociocultural barriers.

Most participants strongly agreed with mental health literacy statements, indicating a good understanding of key concepts. However, during the FGDs, adolescents struggled to define mental health and noted that it is not openly discussed in their communities. Adolescents often struggle to conceptualise mental health, typically framing it negatively using terms like "crazy." Their perspective seems limited to extreme psychiatric disorders, overlooking the spectrum of everyday mental health challenges. Stigma and shame are the reasons why they do not talk about mental health. Adolescents' framing of mental health in a negative way and distancing themselves from those with mental health disorders, "us vs. them," aligns with the notion that stigma may shape how mental health is discussed, rather than an indication of a lack of understanding of the subject. The discrepancy between quantitative and qualitative findings could be an indication that it is easier for this age group to identify mental health when given options to choose from rather than to describe it in their own words. A review by Renwick et al. (2024) found that knowledge about mental illness, treatment and help-seeking was generally low in children and adolescents in LMICs. As reflected in our findings, adolescents

are aware of available mental health services at their local clinics, but avoid them due to negative treatment from clinic staff, which may act as a significant barrier to intervention. Mental health education for adolescents and their caregivers is crucial to help them identify, prevent and manage adolescent mental health, as caregivers are often the primary source of support for adolescents. Low mental health literacy among parents and caregivers can result in adverse implications for adolescents, such as delayed help-seeking and treatment (Gronholm et al., 2015).

In the survey, the majority of participants reported having access to home Wi-Fi or fibre connections, followed by those using mobile data. This suggests that a substantial number of adolescents have internet connectivity. Most participants found internet access easy, but free community Wi-Fi was scarce. Upon further investigation, adolescents explained challenges in accessing free Wi-Fi hotspots, including their distance, overcrowding and the risk of mobile phone theft. Despite participants reporting high mobile data usage in the survey, adolescents mentioned during the FGDs that mobile data costs are high and unaffordable for them. SA data costs are relatively high, and in Gauteng, the percentage of households living below the average poverty line is 36%, up from 25% in 2017/18; Soweto is among the areas affected (Mhlanga and Moloi, 2020; de Kadt et al., 2021). For households under severe resource constraints, mobile data competes with essential expenditures such as food. A study by Kenny et al. (2016) exploring adolescents' perspectives on mental health mobile apps found that adolescents were less likely to use them when required to pay. Adolescents reported that if there is a financial cost, they will not use the apps. Similarly, Maloney et al. (2020) found that users might be unwilling to use their mobile phone data on M-mHealth resources. In 2013, the South African government initiated the South Africa Connect project to provide free Wi-Fi hotspots in major cities. However, these hotspots often experience slow speeds due to high user traffic and imposed daily data limits that are insufficient for users' needs. A notable finding from this research is that the proximity to, and potential theft of, mobile devices deterred participants from accessing the free Wi-Fi hotspots.

Quantitative findings show high smartphone ownership among adolescents, but FGDs revealed that many cannot afford smartphones or data due to financial constraints. Survey results indicated that most adolescents do not share their phones with family members, suggesting they have sole ownership. Furthermore, more than half of them stated that no one in their family can access their phones. Qualitative findings, by contrast, indicated that guardians exercise control by monitoring adolescents' phone use and confiscating phones as punishment. Guardians monitor adolescents' phone use due to concerns about neglect of schoolwork and communication with their boyfriends/girlfriends. This undermines adolescents' autonomy, and they strongly feel that their privacy is being invaded when guardians do this. A study by Chandra et al. (2014) and Duan et al. (2020) piloting M-mHealth interventions for adolescents highlights adolescents' concerns, like sharing devices with parents and parents accessing their mobile devices. For example, in Duan et al. (2020), adolescents reported that parents keep their mobile phones for them most of the day. This limited how they engaged and what they shared on the M-mHealth intervention.

According to Akter et al. (2022), parents are concerned because they do not know what their adolescents do on their mobile phones. Akter et al. (2022) add that parents use parental control apps to monitor their teens' mobile phone use and behaviours to keep them safe. While the use of parental control apps is not popular in the

SA context, guardians still exert this control by directly searching through adolescents' mobile phones or completely taking them away from them. There are many reasons children and adolescents dislike parental control apps, one of which is feeling overly restricted and stripped of autonomy and privacy (Wang et al., 2021). Such restrictions and parental surveillance may impede adolescents' full engagement with M-mHealth interventions. Although M-mHealth is a promising approach for adolescent mental health, numerous pragmatic challenges may hinder its successful implementation and use.

Our findings indicate that adolescents struggle to conceptualise mental health but have a positive attitude towards M-mHealth, as evidenced by both quantitative and qualitative data. One of the key issues identified in a study examining prospects and challenges of M-mHealth interventions in rural South Africa was the lack of mental health education among young people, with only 22% reporting having received any prior education (Mindu et al., 2023). The paradox between the limited conceptualisation of mental health and the enthusiasm for M-mHealth could act as a crucial opportunity to leverage M-mHealth interventions as platforms for promoting mental health literacy. Adolescents' positive attitude could be less driven by mental health itself and more by the general appeal of digital technology. They may like M-mHealth because they are comfortable with mobile devices, because their peers use similar tools or because digital options feel private and provide a sense of control. However, this does not necessarily indicate a good grasp of mental health. If adolescents' understanding of mental health is not strengthened, their use of these tools may end up being superficial, leading to low engagement and high dropout rates.

The quantitative results showed that most respondents would be open to using their mobile phones to address their mental health needs and indicated comfort in sharing their challenges with a professional over the phone. During the FGDs, adolescents highlighted the potential advantages of using M-mHealth, including privacy, convenience and cost savings compared to private mental health care services, which many cannot afford. A digital behavioural activation intervention by Moffett et al. (2025) that uses gamification and peer support for adolescent depression in rural SA demonstrated high acceptability of the Kuamsha app, with additional support from peer mentor calls. These findings suggest that M-mHealth interventions can be effective when they incorporate a hybrid delivery model, rather than relying solely on digital methods. For instance, integrating peer support can enhance the impact of these interventions, as noted by Moffett et al. (2025). Taken together, these findings suggest that adolescents see benefits in M-mHealth but face barriers such as high data costs, limited free Wi-Fi and expensive smartphones. Before adolescents can fully benefit from M-mHealth, these challenges must be addressed so that South African adolescents in communities such as Soweto can also benefit from its potential.

## Conclusion

Our study represents a novel step towards understanding the emerging and rapidly growing field of M-mHealth among adolescents. It also highlighted the importance of using mixed methods to explore this topic. Quantitative findings alone might be misleading, and the nuanced characteristics of these issues require a qualitative approach.

This research makes a unique contribution by highlighting how parental surveillance, device sharing, infrastructural constraints and household dynamics can shape adolescents' engagement with M-mHealth. The findings further show how stigma is a critical barrier that can undermine the uptake and sustained engagement with M-mHealth. Based on the findings, several recommendations emerge. First, co-design should be central to the development of M-mHealth interventions to ensure that they are attuned to specific adolescent populations. Parental engagement is essential for supporting safe engagement and autonomy. Future research should prioritise pilot trials in urban and rural communities, examining supported and standalone approaches and ensuring that they reflect adolescents' priorities. Overall, M-mHealth extends beyond the health sector and is constrained by infrastructural and socio-cultural barriers. Therefore, policymakers need to embed digital M-mHealth strategies within broader youth, education and community development. At the implementation level, strategies should prioritise low-bandwidth, zero-rated and hybrid approaches that combine M-mHealth tools with human support.

## Study limitations

The findings should be considered in light of their small size, with adolescents participating in supportive community programmes from a single youth centre. Only basic demographic data (age and gender) were collected, limiting the ability to account for potentially relevant contextual factors such as socioeconomic background and schooling status. In addition, information on participants' mental health experiences, including diagnoses, prior treatment or family history, was not gathered, which restricted the depth of interpretation. The reliability scores ranged from 0.5 to 0.6 across all constructs, indicating weak reliability. Therefore, the strength of the conclusions drawn from the measures used is limited and should be interpreted with caution. While the TAM framework is valuable, it overlooks socio-cultural and relational factors.

**Open peer review.** To view the open peer review materials for this article, please visit http://doi.org/10.1017/gmh.2026.10154.

**Supplementary material.** The supplementary material for this article can be found at http://doi.org/10.1017/gmh.2026.10154.

**Data availability statement.** The data for this study are available from the lead author upon reasonable request.

**Author contribution.** Conceptualisation: S.M., J.W., L.J.W., M.S.; Data Collection: S.M; Editing: J.W., L.J.W., M.S.; Formal Analysis: S.M.; Methodology: S.M., J.W., L.J.W., M.S.; Supervision: J.W., L.J.W., M.S.; Writing Review: S.M.

**Financial support.** This work was supported by the National Research Foundation of South Africa's PhD funding for S.M. (grant number MND210407592644). J.W. supported by the National Research Foundation of South Africa (grant number 141995). L.J.W. (grant number 301436/Z/23/Z) is supported through a mid-career development award from the Wellcome Trust UK. SAMRC-Wits DPHRU is supported by the South African Medical Research Council.

**Competing interests.** The authors declare that they have no competing interests in the publication of this article.

**Ethics statement.** Ethical clearance was obtained from the University of the Witwatersrand's Human Research Ethics Committee (clearance number H22/10/30).

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
