## [Reviewer Report]

Dear Authors,

Thank you for the opportunity to review this manuscript. This is a timely and important study exploring adolescents’ perceptions of mobile mental health in Soweto. The mixed-methods design is well chosen, and the use of the Pillar Integration Process is a methodological strength. The paper provides valuable qualitative insights into stigma and parental dynamics, and the focus on autonomy versus ownership of phones is a novel and important contribution.

That said, the manuscript requires substantial revision. In particular, the Abstract and Introduction need sharper framing, stronger engagement with existing literature (especially South African work and systematic reviews of mental health apps), and clearer articulation of aims. The Methods need fuller reporting of ethics approval, exclusion criteria, and the focus group guide; measurement reliability issues must also be acknowledged. The Results are strong but could more directly highlight contradictions (e.g., limited conceptualisation of mental health yet enthusiasm for digital interventions), stratify findings, and emphasise the striking data on clinic stigma. The Discussion and Conclusion are currently underdeveloped, relying too much on “more research is needed” rather than offering clear implications or recommendations.

Below I outline detailed comments, grouped by section.

Abstract

1. The opening line focuses on undiagnosed disorders. In this context, the bigger issue is untreated disorders, which persist into adulthood.

2. Barriers are framed narrowly as lack of workers, infrastructure, and medication. The broader problem is access to treatment and support, including stigma, privacy, and affordability.

3. The phrase “struggle to conceptualise mental health” is vague. Clarify whether this means adolescents do not understand the construct of mental health or cannot articulate what makes mental health good/bad.

4. Stigma is a central finding (e.g., “crazy,” “bewitched”), but not mentioned in the Abstract. This should be included.

5. The conclusion is too generic (“promising but with challenges”). A stronger summary would highlight specific barriers such as privacy, surveillance, stigma, and data costs. Suggested reframe: “While adolescents were positive about M-mHealth, significant barriers including privacy, high data costs, and stigma highlight that digital tools are not a quick fix. Youth perspectives are critical before scaling digital mental health in South Africa.”

Introduction

6. The opening is too broad (general mHealth examples). It would be stronger to start directly with mental health, where there is a large and growing literature.

7. The introduction lacks critical evaluation of the M-mHealth literature. Reviews show mixed efficacy, poor sustained engagement, and rollout challenges. This must be acknowledged.

8. The paper does not discuss how M-mHealth might be delivered (standalone, therapist-supported, hybrid). This matters for feasibility.

9. The novelty is not well articulated. The contribution of this Soweto study needs clearer positioning.

10. The study’s aims/objectives are not clearly stated. Based on the design, there are two: (i) to assess adolescents’ mental health literacy, and (ii) to explore their perceptions and intentions regarding M-mHealth. Suggested text: “Accordingly, this study had two main objectives: (i) to assess adolescents’ mental health literacy in Soweto, and (ii) to explore their perceptions, attitudes, and intentions regarding mobile mental health (M-mHealth) interventions.”

11. Both Introduction and Discussion underrepresent South African literature. Important omissions include, for example:

• Mindu, T., Mutero, I. T., Ngcobo, W. B., Musesengwa, R., & Chimbari, M. J. (2023). Digital mental health interventions for young people in rural South Africa: Prospects and challenges for implementation. International Journal of Environmental Research and Public Health, 20(2), 1453.

• Moffett, B. D., Pozuelo, J. R., Musenge, E., Makhanya, Z., O’Mahen, H. A., Craske, M. G., ... & DoBAt & Ebikolwa Consortium. A gamified digital intervention using behavioural activation for adolescent depression in rural South Africa: A pilot randomised controlled trial (the DoBAt Study).

• Pozuelo, J. R., Moffett, B. D., Davis, M., Stein, A., Cohen, H., Craske, M. G., ... & O’Mahen, H. A. (2023). User-centered design of a gamified mental health app for adolescents in sub-Saharan Africa: Multicycle usability testing study. JMIR Formative Research, 7, e51423.

12. There is also a growing literature specifically on mental health apps’ feasibility, acceptability and implementation. For example, Bear et al. (2022) systematically reviewed markers of successful implementation of youth mental health apps, and Bear et al. (2024) reviewed the acceptability, engagement, and feasibility of such apps for underserved young people. Both reviews emphasise that while acceptability is generally high, sustained engagement is poor, and challenges around adaptation, co-design, and feasibility in low-resource or underserved groups are key. These themes resonate strongly with your findings (privacy, cost, stigma, parental surveillance), and bringing this literature into the Introduction and Discussion would strengthen the framing considerably.

• Bear, H. A., Ayala Nunes, L., DeJesus, J., Liverpool, S., Moltrecht, B., Neelakantan, L., ... & Fazel, M. (2022). Determination of markers of successful implementation of mental health apps for young people: Systematic review. Journal of Medical Internet Research, 24(11), e40347.

• Bear, H. A., Ayala Nunes, L., Ramos, G., Manchanda, T., Fernandes, B., Chabursky, S., ... & Fazel, M. (2024). The acceptability, engagement, and feasibility of mental health apps for marginalized and underserved young people: Systematic review and qualitative study. Journal of Medical Internet Research, 26, e48964.

Methods

13. The mixed-methods design and description of PIP are strong and add credibility. All participants came from a single Soweto youth centre — acknowledgment of this in the Discussion is encouraged.

14. Measurement reliability: Cronbach’s alphas for TAM constructs were 0.5–0.6, which is weak. This undermines the strength of conclusions drawn from these scales and should be acknowledged.

15. Ethics: Guardian consent and adolescent assent are described, but there is no clear statement of formal ethics approval (committee and reference number). If this appears on the title page, it should also be explicit in the Methods.

16. Eligibility criteria are described (age 13–19, Soweto, youth centre), but exclusion criteria are not specified.

17. The FGD topic guide is not included (nor in the supplement). This should be provided for transparency.

18. Demographic data are limited (age, gender). Socioeconomic background and schooling status would have provided crucial context. Participants’ own mental health experiences (diagnoses, treatment, family history) are also not reported. This limits interpretation of their views and should at least be acknowledged.

19. The paper uses TAM for technology uptake and the Mental Health Literacy Measure for knowledge, which are appropriate. However, the study does not employ a broader theoretical framework for adolescents’ beliefs about mental health and treatment. Several models could have been useful, for example:

• Illness Perceptions Model (Common-Sense Model): could frame how adolescents understand identity, causes, consequences, and controllability of mental health.

• Health Belief Model (HBM): could map perceived susceptibility, severity, benefits, and barriers.

• Theory of Planned Behavior (TPB): could explain how attitudes, norms, and control shape behavioural intentions.

• Kleinman’s Explanatory Models: could capture cultural idioms and meanings (“crazy,” “bewitched”).

Bringing in one of these models would not require re-analysis but could provide a clearer theoretical anchor, moving the findings beyond description. At minimum, acknowledging this as a limitation and suggestion for future work would strengthen the paper.

Results

20. Phone ownership and internet access data are highly relevant. The unpacking of ownership vs. autonomy (sharing, surveillance, confiscation) is a novel and important contribution.

21. There seems to be a contradiction between adolescents’ limited conceptualisation of mental health and their strong enthusiasm for M-mHealth. This paradox should be highlighted.

22. Results drawn from weak scales (alphas 0.5–0.6) are presented without caution.

23. No socioeconomic context is provided to interpret barriers like data cost.

24. Subgroup analysis (e.g., gender, age) is not presented but could yield important insights.

25. Qualitative findings on clinic stigma and negative staff treatment are powerful but underemphasised.

Discussion / Conclusion

26. The citation of an Ireland app prototype study (Kenny et al.) feels out of place. SA/LMIC studies should be prioritised. South African literature (DoBAt, Mindu, Khanya) is underrepresented.

27. The paradox between low mental health literacy and enthusiasm for M-mHealth is not adequately unpacked.

28. The paper does not consider how different delivery models (standalone, supported, hybrid) might alter feasibility in this context.

29. There is no dedicated limitations section. This should acknowledge: single-site sample, modest size, non-representativeness, weak scale reliability, and lack of MH experience data.

30. The Conclusion is too generic (“more research is needed”). It should instead:

• Emphasise the unique contributions (autonomy vs. ownership, stigma, parental surveillance).

• Offer specific recommendations (youth co-design, affordability solutions, parental engagement, stigma reduction).

• Suggest concrete research directions (pilot trials in SA, supported vs. standalone models, rural vs. urban comparisons).

In summary: This paper addresses a critical question and has several strengths, especially its methodological design and its novel contribution on autonomy versus ownership. However, revisions are needed to strengthen the framing, better contextualise the findings within South African and global literature, acknowledge study limitations, and expand the recommendations.

---

## [Reviewer Report]

This manuscript explores South African adolescents’ perceptions, attitudes, and intentions to use mobile mental health interventions, using an explanatory sequential mixed-methods design. The topic is timely and important given the global push for scalable digital mental health solutions in LMICs. The study’s strength lies in its contextual grounding in Soweto and its thoughtful integration of methods. However, conceptual clarity, analytical depth, and theoretical framing need further refinement to meet the standards of a high-impact journal.

Major comments

1.     The paper applies the Technology Acceptance Model (TAM) but does not adequately justify its suitability for adolescents in LMICs. TAM’s focus on perceived usefulness and ease of use may overlook socio-cultural and relational factors—such as parental surveillance, stigma, and digital inequities—that shape adolescents’ readiness.

2.     The authors should articulate what “readiness” entails—technological access, digital literacy, attitudinal openness, or socio-cultural acceptability. At present, it is treated as an implicit construct.

3.     The conclusion that “M-mHealth might not be relevant at this point” is overstated. The evidence points more to conditional readiness constrained by infrastructural and socio-cultural barriers rather than conceptual unpreparedness among adolescents.

4.     Reported reliability scores are low (Cronbach α < 0.6) and should be discussed explicitly in terms of implications for validity.

5.     Sampling from one youth centre limits generalisability and may introduce bias (e.g., adolescents already engaged in supportive programmes). This needs to be acknowledged and discussed in terms of implications for validity.

6.     Clarify how saturation was reached and how coding reliability was ensured.

7.     The discussion should interpret findings rather than repeat results.

8.     Clarify the phrase ‘go back to basics’ and specify concrete implications such as digital literacy, zero-rated apps, and co-design with adolescents.

9.     Include discussion on adolescent privacy, digital ethics, and data protection in mHealth interventions.

10.  Include a methodological strengths and limitations subsection in the discussion addressing some of the issues mentioned above, with emphasis on reliability, transferability, and researcher positionality.

11.  The discussion would benefit from more explicit policy and implementation implications of the findings.

Minor comments

1.     Reduce redundancy across sections and proofread for minor grammatical issues.

2.     Clarify Ns and percentages in tables where totals do not align

3.     Consider shortening the title to ‘Adolescent readiness for mobile mental health support in Soweto: A mixed-methods study.’

---

## [Editor Report]

Dear Authors 

We have now received comments from reviewers on your manuscript. In light of those comments, we recommend major revisions to be made to your current manuscript, for it to be considered for publication. 

Regards

Siham

---

## [Reviewer Report]

I would like to thank the authors for their careful and detailed response to my initial review. The manuscript has improved substantially in its framing, engagement with South African and LMIC literature, transparency regarding methodological limitations, and interpretation of findings. Overall, the authors have addressed the vast majority of my substantive concerns, and the paper is close to being suitable for publication. The remaining issues are minor but important for conceptual clarity, internal consistency, and polish before acceptance.

Remaining points to address

1. Definition and use of “readiness”

The concept of “readiness” remains central to the paper but is still used somewhat implicitly. At present, it appears to encompass a combination of attitudinal openness, technological access, and socio-cultural constraints. I recommend adding a brief clarifying statement (1–3 sentences) explicitly defining how “readiness” is conceptualised in this study.

2. Definition of mHealth / M-mHealth in the Introduction

Although mobile mental health (M-mHealth) is clearly defined in the abstract, it would improve accessibility to spell out mobile health (mHealth) and mobile mental health (M-mHealth) in full at their first mention in the Introduction, rather than relying on the abstract definition.

3. Redundancy in the Introduction

Following revision, there is some repetition in the Introduction. In particular, the statistic that more than one in ten children in South Africa has a diagnosable and treatable mental health disorder appears in more than one paragraph. Consolidating this statistic into a single location would improve narrative flow and reduce redundancy.

4. Interpretation of quantitative findings given weak reliability

The authors appropriately acknowledge that several survey scales show weak internal consistency (Cronbach’s α ≈ 0.5–0.6). However, some statements in the Results and Discussion still describe quantitative findings in relatively strong terms (e.g., “most participants agreed”) without consistent qualification. I recommend ensuring that quantitative findings are consistently framed as exploratory or descriptive, and that interpretation clearly prioritises the qualitative findings where scale reliability is limited.

5. Delivery models (standalone vs supported/hybrid)

The discussion of hybrid or supported delivery models is helpful, but at times reads more strongly than warranted by the data. As delivery models were not directly examined in this study, I recommend reframing these points explicitly as implications inferred from the findings (e.g., privacy concerns, clinic stigma, parental surveillance), rather than as conclusions drawn from direct evidence.

6. Theoretical framing remains largely descriptive

The authors appropriately acknowledge as a limitation that broader theoretical frameworks (e.g., Kleinman’s explanatory models, HBM, TPB) were not employed. However, the discussion would benefit from a short illustrative paragraph mapping one or two key findings (e.g., stigma, “crazy/bewitched” idioms, parental surveillance) onto a relevant framework, even if only interpretively. This would move the discussion slightly beyond description without requiring re-analysis.

7. Minor editorial tightening

The Discussion is substantially improved but remains dense in places, with some long, multi-purpose paragraphs. Light tightening or paragraph splitting would further improve readability and reduce residual repetition across sections.

Subject to these minor clarifications and editorial refinements, I am satisfied that the authors have addressed my previous concerns and strengthened the manuscript considerably. I would support acceptance pending minor revisions.

---

## [Editor Report]

Dear Authors 

Thanks for addressing the reviewer’s comments and submitting the revised manuscript, we are pleased to inform you that it is accepted for publication. We will follow up with next steps. 

Warmly

Siham